# Identification and comparison of orthologous cell types from primate embryoid bodies shows limits of marker gene transferability

Jessica Jocher[1†], Philipp Janssen[1†], Beate Vieth[1], Fiona C Edenhofer[1], Tamina Dietl[2], Anita Térmeg[1], Paulina Spurk[1], Johanna Geuder[1], Wolfgang Enard[1*‡], Ines Hellmann[1*‡]

[1]Anthropology and Human Genomics, Faculty of Biology, Ludwig-Maximilians-Universität München, Munich, Germany; [2]Helmholtz Zentrum München Deutsches Forschungszentrum für Gesundheit und Umwelt: Munich, Munich, Germany

**\*For correspondence:**
enard@bio.lmu.de (WE);
hellmann@bio.lmu.de (IH)

[†]These authors contributed equally to this work
[‡]These authors also contributed equally to this work

## eLife Assessment

The authors make an **important** contribution to comparative functional genomics by developing a semi-automated computational pipeline that integrates classification and marker-based cluster annotation to identify orthologous cell types. Using a single-cell RNA-seq dataset of induced pluripotent stem cells and derived embryonic bodies from four primate species: humans, orangutans, cynomolgus macaques, and rhesus macaques, the authors provide **convincing** evidence that cell type-specific marker genes are substantially less transferable across species than broadly expressed genes, with transferability declining as phylogenetic distance increases. This study establishes a key framework and reference dataset for comparative single-cell analyses and encourages more rigorous evaluation of marker gene transferability across species.

**Abstract** The identification of cell types remains a major challenge. Even after a decade of single-cell RNA sequencing (scRNA-seq), reasonable cell type annotations almost always include manual non-automated steps. The identification of orthologous cell types across species complicates matters even more, but at the same time strengthens the confidence in the assignment. Here, we generate and analyze a dataset consisting of embryoid bodies (EBs) derived from induced pluripotent stem cells (iPSCs) of four primate species: humans, orangutans, cynomolgus, and rhesus macaques. This kind of data includes a continuum of developmental cell types, multiple batch effects (i.e. species and individuals) and uneven cell type compositions and hence poses many challenges. We developed a semi-automated computational pipeline combining classification and marker-based cluster annotation to identify orthologous cell types across primates. This approach enabled the investigation of cross-species conservation of gene expression. Consistent with previous studies, our data confirm that broadly expressed genes are more conserved than cell type-specific genes, raising the question of how conserved, inherently cell type-specific, marker genes are. Our analyses reveal that human marker genes are less effective in macaques and vice versa, highlighting the limited transferability of markers across species. Overall, our study advances the identification of orthologous cell types across species, provides a well-curated cell type reference for future in vitro studies and informs the transferability of marker genes across species.

## Introduction

Cell types are a central concept for biology, but are, as other concepts like species, practically difficult to identify. Theoretically, one would consider all stable, irreversible states on a directed developmental trajectory as cell types. In practice, we are limited by our experimental possibilities. Historically, cell type definitions hinged on observations of cell morphology in a tissue context, which was later combined with immunofluorescence analyses of marker genes (*Bakken et al., 2017*). A lot of the functional knowledge that we have about cell types today is based on such visual and marker-based cell type definitions. With single-cell sequencing, our capabilities to characterize and identify new cell types have radically changed (*The Tabula Muris Consortium et al., 2018*; *Regev et al., 2017*). Clustering cells by their expression profiles enables a more systematic and higher-resolution identification of groups of cells that are then interpreted as cell types. However, distinguishing them from cell states or technical artifacts is not straightforward. A key criterion for defining a true cell type is its reproducibility across experiments, individuals, or even species.

Hence, identifying the same, i.e., orthologous, cell types across individuals and species is crucial. There are three principal strategies to match cell types from scRNA-seq data. (1) One is to integrate all cells prior to performing a cell type assignment on a shared embedding (*Song et al., 2023*). (2) The second approach is to consider cell types from one species as the reference and transfer these annotations to the other species using classification methods (*Liu et al., 2023*). (3) The third strategy is to assign clusters and match them across species, which has the advantage of not requiring data integration of multiple species or an annotated reference (*Castro-Mondragon et al., 2022*; *Bakken et al., 2021*; *Suresh et al., 2023*).

Furthermore, established marker genes are still heavily used to validate and interpret clusters identified by scRNA-seq data (*Zhang et al., 2019b*; *Guo and Li, 2021*; *Ianevski et al., 2022*). Together with newly identified transcriptomic markers for human and mouse, they are collected in databases (*Franzén et al., 2019*; *Zhang et al., 2019a*) and provide the basis for follow-up studies using spatial transcriptomics and/or immunofluorescence approaches. However, previous studies have shown that the same cell types may be defined by different marker genes in different species (*Hodge et al., 2019*; *Bakken et al., 2021*). For example, *Krienen et al., 2020* found that only a modest fraction of interneuron subtype-specific genes overlapped between primates and even less between primate and rodent species.

To better understand how gene expression in general and the expression of marker genes in particular evolves across closely related species, we used induced pluripotent stem cells (iPSCs) and their derived cell types from humans and non-human primates (NHP). One fairly straightforward way to obtain diverse cell types from iPSCs are embryoid bodies (EBs). EBs are the simplest type of iPSC-derived organoids contain a dynamic mix of cell types from all three germ layers and result from spontaneous differentiation upon withdrawal of key pluripotency factors (*Brickman and Serup, 2017*; *Itskovitz-Eldor et al., 2000*; *Rhodes et al., 2022*; *Guo et al., 2019*; *Han et al., 2018*).

EBs and brain organoids from humans and chimpanzees have, for example, been used to infer human-specific gene regulation in brain organoids (*Kanton et al., 2019*) or to investigate mechanisms of gene expression evolution (*Barr et al., 2023*).

Here, we explore to what extent levels of cell type specificity of marker genes are conserved in primates. We generated scRNA-seq data of 8 and 16-day-old EBs from human, orangutan (*Pongo abelii*), cynomolgus (*Macaca fascicularis*), and rhesus macaque (*Macaca mulatta*) iPSCs. Using this data, we established an analysis pipeline to identify and assign orthologous cell types. With this annotation, we provide a well-curated cell type reference for in vitro studies of early primate development. Moreover, it allowed us to assess the cell type specificity and expression conservation of genes across species. We find that even though the cell type-specificity of a marker gene remains similar across species, its discriminatory power still decreases with phylogenetic distance.

## Results

### Generation of embryoid bodies from iPSCs of different primate species

We generated EBs from iPSCs across multiple primate species: two human iPSC clones (from two individuals), two orangutan clones (from one individual), three cynomolgus clones (from two individuals), and three rhesus clones (from one individual) (*Geuder et al., 2021*; *Jocher et al., 2024*;

*Edenhofer et al., 2024*). To optimize conditions for generating a sufficient number of cells from all three germ layers across these four species, we tested combinations of two culturing media ('EB-medium' and 'DFK20,' see Methods) and two EB-differentiation conditions ('single-cell seeding' and 'clump seeding,' see Methods). After 7 days of differentiation, germ layer composition was analyzed by flow cytometry (*Figure 1—figure supplement 1A, B and C*). Among the four tested protocols, culture in DFK20 medium with clump seeding resulted in the most balanced representation of all germ layers, yielding a substantial number of cells from each layer across all species (*Figure 1—figure supplement 1D*).

Under these conditions, we established an EB formation protocol based on 8 days of floating culture in dishes, followed by 8 days of attached culture (*Figure 1A*). This results in the formation of cells from all three germ layers, as confirmed by immunofluorescence staining for AFP (endoderm), β-III-tubulin (ectoderm) and α-SMA (mesoderm) (*Figure 1B*). To generate scRNA-seq data, we dissociated 8 or 16-day-old EBs into single cells and pooled cells from all four species to minimize batch effects (*Figure 1C*). We performed the experiment in three independent replicates, generating a total of four lanes and six lanes of 10 x Genomics scRNA-seq at day 8 and day 16, respectively (*Figure 1—figure supplement 2A*). This resulted in a dataset comprising over 85,000 cells after filtering and doublet removal, distributed fairly equally over time points, species, and clones (*Figure 1—figure supplement 2B–D*).

In agreement with the immunofluorescence staining, we detected well-established marker genes of pluripotent cells and of all three germ layers (*Ludwig et al., 2023*) in the scRNA-seq data: *SOX2*, *SOX10*, and *STMN4* expression was used to label ectodermal cells, *APOA1,* and *EPCAM* for endodermal cells, *COL1A1* and *ACTA2* (α-SMA) for mesodermal cells, and *POU5F1* and *NANOG* for pluripotent cells (*Figure 1D*). Expression of these marker genes corresponded well with a classification based on a published scRNA-seq dataset from 21-day-old human EB (*Rhodes et al., 2022*). This initial, rough germ layer assignment shows that our differentiation protocol generates EBs with the expected germ layers and cell type diversity from all four species (*Figure 1E*, *Figure 1—figure supplement 3A*).

## Assignment of orthologous cell types

Many integration methods encounter difficulties when they are applied to data from multiple species and uneven cell type compositions (*Song et al., 2023*). Indeed, when comparing clusters derived from an integrated embedding across all species (*Hie et al., 2019*; *Korsunsky et al., 2019*) to the aforementioned preliminary cell type assignments, we observed signs of overfitting. For instance, a cluster predominantly containing cells classified as neurons in humans, cynomolgus, and rhesus macaques consisted mainly of early ectoderm and mesoderm cells in orangutans (*Figure 1—figure supplement 3B and C*). To address this issue, we developed an approach that assigns orthologous cell types without a common embedding space in an interactive shiny app (https://shiny.bio.lmu.de/Cross_Species_CellType/; *Figure 2A and B*):

First, we assign cells to clusters separately for each species. To avoid losing rare cell types, we aim to obtain at least double as many high-resolution clusters (HRCs) per species as expected cell types. We then use the HRCs of one species as a reference to classify the cells of the other species using SingleR (*Aran et al., 2019*). These pair-wise comparisons are done reciprocally for each species and via a cross-validation approach also within each species (see Methods). For each comparison, we average the two values for the fraction of cells annotated as the other HRC. For example, a perfect 'reciprocal best-hit' between HRC-A in human and HRC-B in rhesus would have all cells of HRC-B assigned to HRC-A when using the human as a reference and reciprocally all cells in HRC-A assigned to HRC-B when using the rhesus as a reference. Next, we used the resulting distance matrix as input for hierarchical clustering to find orthologous clusters across species and merge similar clusters within species. Here, the user can choose and adjust the final cell type cluster number. This allows us to identify orthologous cell type clusters (OCCs) across all four species, while retaining species-specific clusters when no matching cluster was identified.

In the last steps, OCCs are manually further refined by merging neighboring OCCs with similar marker gene and transcriptome profiles (see Methods). To avoid bias, we first identify marker genes independently for each species solely based on scRNA-seq expression data (*Hao et al., 2021*). We then intersect those lists to identify the top-ranking marker genes with consistently good specificity across all species. The final set of conserved marker genes then serves us to derive cell type labels by

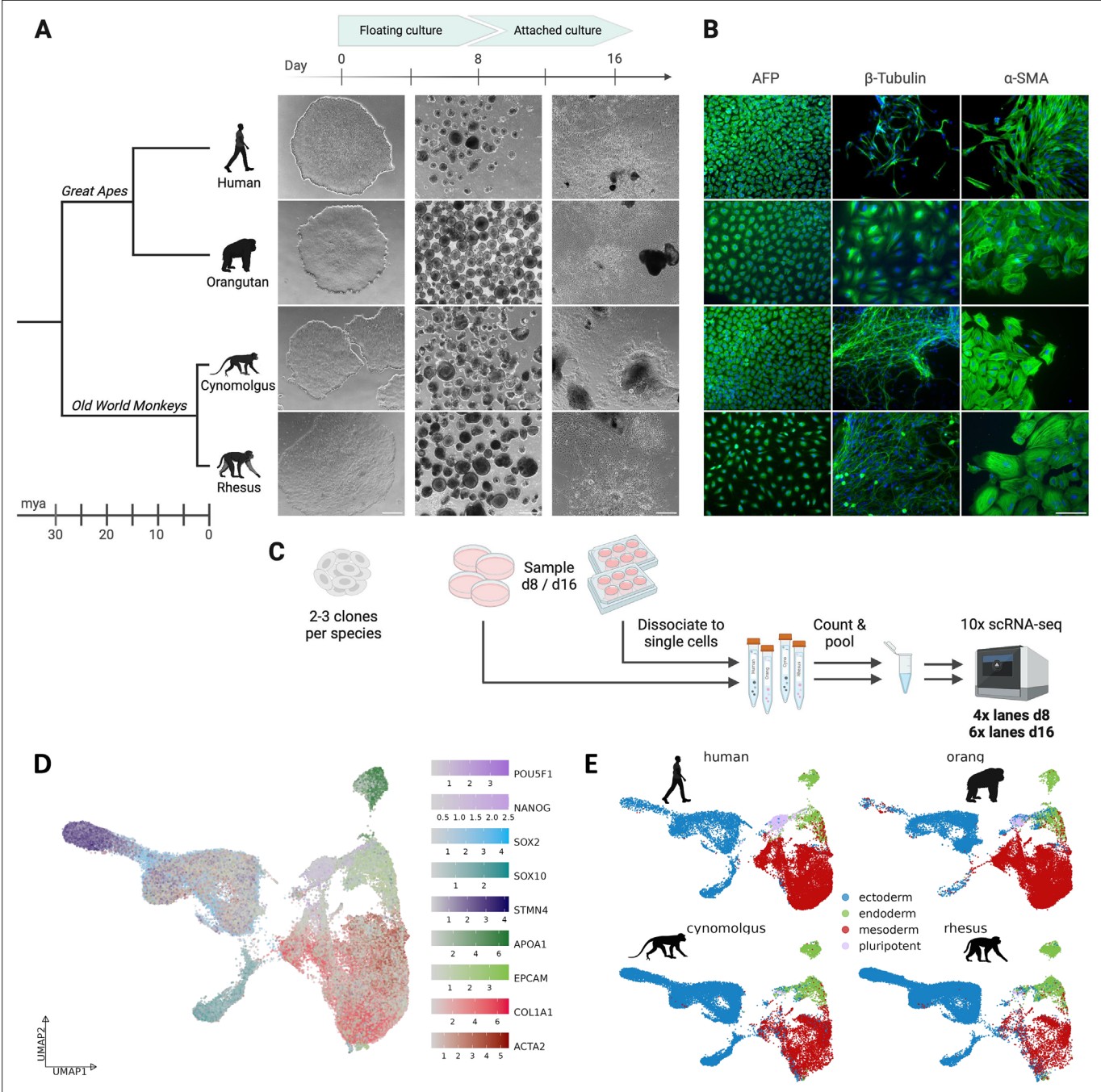

**Figure 1.** Generation of primate embryoid bodies. (**A**) Overview of the embryoid body (EB) differentiation workflow of the four primate species human (*Homo sapiens*), orangutan (*Pongo abelii*), cynomolgus (*Macaca fascicularis*), and rhesus (*Macaca mulatta*), including their phylogenetic relationship. Scale bar represents 500 µm. (**B**) Immunofluorescence staining of day 16 EBs using $\alpha$-fetoprotein (AFP), $\beta$-III-tubulin, and $\alpha$-smooth muscle actin ($\alpha$-SMA). Scale bar represents 100 µm. (**C**) Schematic overview of the sampling and processing steps prior to 10 x scRNA-seq. (**D**) UMAP representation of the whole scRNA-seq dataset, integrated across all four species with Harmony. Single cells are colored by the expression of known marker genes for the three germ layers and undifferentiated cells. (**E**) UMAP representation, colored by assigned germ layers, split by species. Created with BioRender.com.

The online version of this article includes the following figure supplement(s) for figure 1:

**Figure supplement 1.** Comparison of embryoid body (EB) differentiation protocols using flow cytometry.

**Figure supplement 2.** Total number of recovered cells.

**Figure supplement 3.** Reference-based cell type classification.

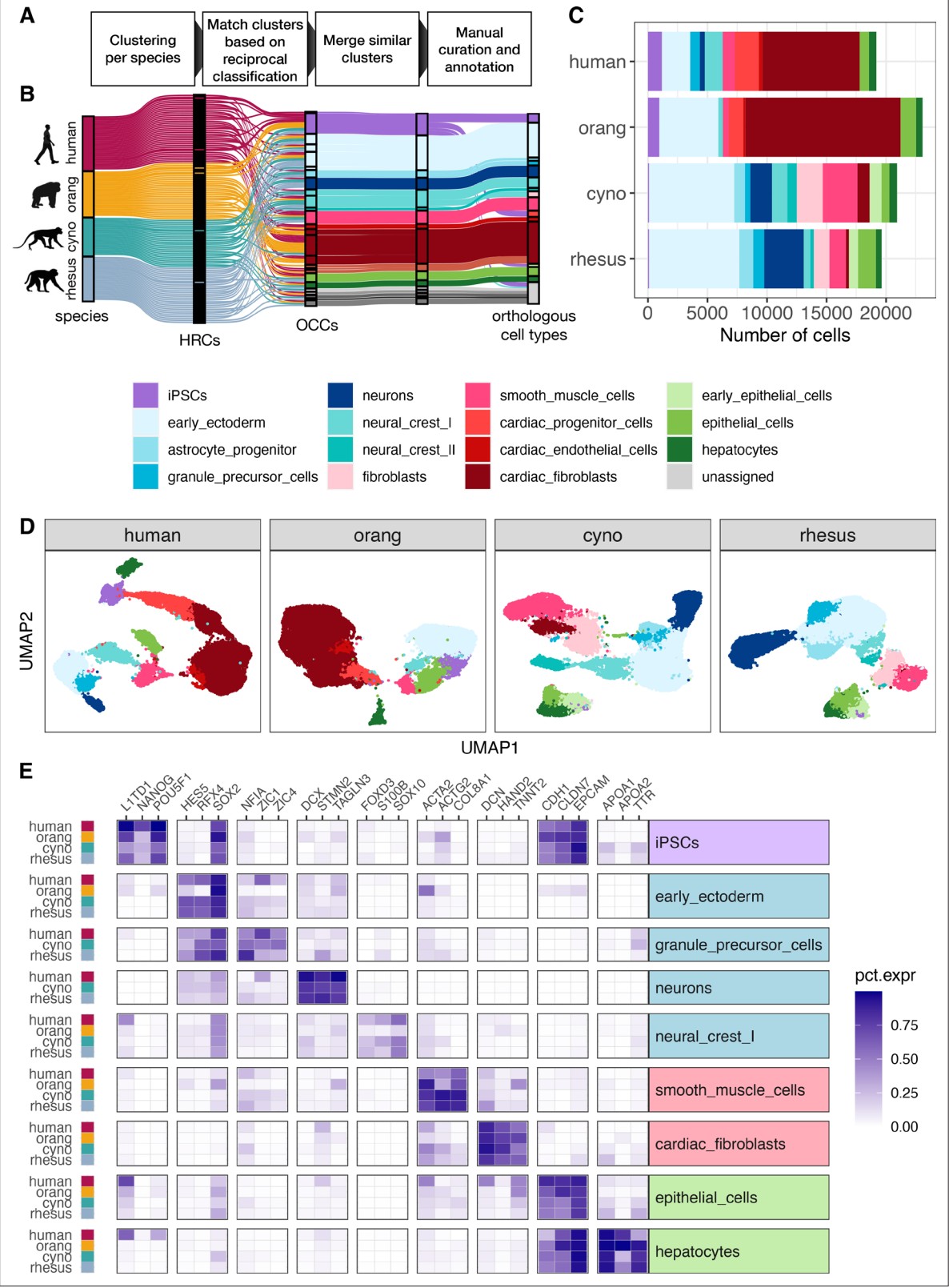

**Figure 2.** Assignment of orthologous cell types across species. (**A**) Schematic overview of the pipeline to match clusters between species and assign orthologous cell types. (**B**) Sankey plot visualizing the intermediate steps of the cell type assignment pipeline. Each line represents a cell which are colored by their species of origin on the left and by their current cell type assignment during the annotation procedure on the right. An initial set of 118 high-resolution clusters (HRCs), 25–35 per species, was combined into 26 orthologous cell type clusters (OCCs). Similar cell type clusters were merged,

*Figure 2 continued on next page*

*Figure 2 continued*

and after further manual refinement, provided the basis for final orthologous cell type assignments. (C) Fraction of annotated cell types per species. (D) UMAPs for each species colored by cell type. (E) To validate our cell type assignments, we selected three marker genes per cell type that exhibit a similar expression pattern across all four species and have been reported to be specific for this cell type in both human and mouse (*Appendix 1—table 1*). The heatmap depicts the fraction of cells of a cell type in which the respective gene was detected for cell types present in at least three species.

The online version of this article includes the following figure supplement(s) for figure 2:

**Figure supplement 1.** Replicability of cell types across species measured by reciprocal classification.

**Figure supplement 2.** Replicability of cell types across species measured with MetaNeighbor.

**Figure supplement 3.** Cell type annotation.

**Figure supplement 4.** Pseudotime analysis of ectoderm differentiation trajectories.

**Figure supplement 5.** Pseudotime analysis of mesoderm differentiation trajectories.

**Figure supplement 6.** Pseudotime analysis of endoderm differentiation trajectories.

searching the literature as well as databases of known marker genes (*Figure 2E*). If the marker-gene-based cell type assignment reveals cluster inconsistencies, they can be marked for further splitting. This feature is of particular importance for rare cell types. For example, we separated a cluster of early progenitor cells into iPSCs, cardiac progenitors, and early epithelial cells.

*Suresh et al., 2023* devised a conceptually similar approach to ours to identify orthologous cell types across species. The main difference is that they used scores from MetaNeighbor *Crow et al., 2018* where we use SingleR to measure distances between HRCs. However, in essence, both scores are based on rank correlations, and hence it may not be surprising that both scoring systems yield consistent cluster groupings that show high replicability across species. However, using our SingleR-based scores to compare OCCs across species may yield more clearly defined correspondences compared to MetaNeighbor scores (*Figure 2—figure supplements 1 and 2*).

Overall, we are confident that our approach yields meaningful orthologous cell type assignments, without requiring a prior annotation per species or a reference dataset. Moreover, the necessary fine-tuning of the cell type clusters by the expert user is facilitated by an interactive app.

## Many cell types are shared between day 8 and day 16 EBs

Using the strategy described in the previous section, we detected a total of 15 reproducible cell types from the three germ layers, all of which were detected in at least three cell lines in three independent replicates. Among these, we identified four cell types that represent the latest time points along ectodermal developmental lineages (astrocyte progenitor, granule precursor, neurons, neural crest II), four that represent the latest time points along mesodermal lineages (fibroblasts, smooth muscle cells, cardiac endothelial cells, cardiac fibroblasts), and two that represent the latest detected time points along endodermal lineages (epithelial cells, hepatocytes). Many of these cell types were present at both sampling times (*Figure 2—figure supplement 3C*). The most notable exception is that orangutan EBs lost the majority of ectodermal cells at the later time point. Aside from this technical deviation—likely caused by the additional handling step (see previous chapter)—some more differentiated cell types only appear at day 16 at appreciable frequencies. This is most pronounced for smooth muscle cells in all species, but also holds for neuron-like cells in humans. Overall, this leads to an increase in the observed cell type diversity over time.

To further evaluate differences between the two sampling time points, we performed pseudotime analyses (*Street et al., 2018*) on the experiments integrated per species and germ layer, defining iPSCs as the origin and the differentiated cell types listed above as the endpoints of the developmental trajectories (*Figure 2—figure supplements 4–6*). As expected, day 16 cells generally occupy later positions along the trajectories than day 8 cells, yet the distributions overlap: iPSCs and precursor states, such as early ectoderm are still detectable, albeit at lower frequency, in the day 16 EBs. Still, the few states that are confined to one of the two time points improve cross-species comparability when both are considered jointly. Integrating day 8 and day 16 increased the overlap in detected cell types between species; for example, human neural cells were only observed at day 16, whereas they were already present at day 8 in macaques, and we, therefore, used the combined data from both time points for downstream analyses.

Overall, 9 of the 15 cell types were detected in at least 3 species, and 7 cell types were reproducibly detected in all four species (*Figure 2C and D*; *Figure 2—figure supplement 3*). These 7 cell types consisted of iPSCs, two cell types representing ectoderm: early ectoderm and neural crest, two cell types of mesodermal origin: smooth muscle cells and cardiac fibroblasts, and two endodermal cell types: epithelial cells and hepatocytes (*Figure 2C and E*) and are used for the analysis of pleiotropy and marker genes in the remainder of this manuscript.

## Cell type-specific genes have less conserved expression levels

Based on the premise that it is not necessarily the expression level, but rather the expression breadth that determines expression conservation (*Duret and Mouchiroud, 2000*), we developed a method to call a gene 'expressed' or not that considers the expression variance across the cells of one type, which we then used to score cell type-specificity and expression conservation (*Figure 3B*); see Methods.

For example, we find that the neural crest marker *SOX10* (*Mollaaghababa and Pavan, 2003*) is cell type-specific and conserved, the lncRNA *ESRG* is iPSC- and human-specific; in contrast, *RPL22*, a gene that encodes a protein of the large ribosomal subunit, is broadly expressed and conserved (*Figure 3A*). Overall, we find on average ~15% of genes to be cell type-specific, i.e., our score determined them to be expressed in only one cell type, while ~40% of genes were found to be broadly expressed in all seven cell types (*Figure 3—figure supplement 1A*).

Additionally, we obtained a measure of expression conservation, which quantifies the consistency of the cell type expression score across species. We found that broadly expressed genes present in all cell types exhibited high expression conservation, whereas cell type-specific genes tended to be more species-specific (*Figure 3C*; *Figure 3—figure supplement 1B*).

Unsurprisingly, broadly expressed genes also showed higher average expression levels (*Kliesmete et al., 2024*; *Figure 3—figure supplement 1D*). To ensure that the observed relationship between expression breadth and conservation in our data is not solely due to expression level differences, we sub-sampled genes from all cell type-specificity levels for comparable mean expression. This did not change the pattern: also, broadly expressed genes with a low mean expression level are highly conserved across species (*Figure 3—figure supplement 1E and F*). Moreover, the coding sequences of broadly expressed genes show higher levels of constraint than more cell type-specific genes, thus supporting the notion that the higher conservation of the expression pattern that we observed here is due to evolutionary stable functional constraints on this set of genes (*Figure 3D*; *Figure 3—figure supplement 1C*).

## Marker gene conservation

Building on our previous observation that cell type-specific genes are less conserved across species, we investigated the conservation and transferability of marker genes, which are, by definition, cell type-specific, in greater detail. To this end, we call marker genes for all cell types and species, using a combination of differential expression analysis and a quantile rank-score based test for differential distribution detection (*Ling et al., 2021*). Additionally, we define a good marker gene as one that is upregulated and expressed in a higher fraction of cells compared to the rest. To prioritize marker genes, we rank them based on the difference in the detection fraction: the proportion of cells of a given type in which a gene is detected compared to its detection rate in all other cells.

We found a low overlap of top marker genes among species, with a median of 15 of the top 100-ranked marker genes per cell type shared across all four species, while a larger proportion of markers was unique to individual species (*Figure 4A*). Notably, these species-specific markers often exhibited cell type-specific expression in only one species, with reduced or non-specific expression in others (*Figure 4B*; *Figure 4—figure supplement 1*).

Given the special role of transcriptional regulators for the definition of a cell type (*Arendt et al., 2016*) and the differences in conservation between protein-coding and non-coding RNAs (*Johnsson et al., 2014*), we analyzed the comparability of marker genes of different types. To this end, we assessed the concordance of the top 100 marker genes across species for protein-coding genes, lncRNAs, transcription factors (TFs), or all genes using rank-biased overlap (RBO) scores (*Webber et al., 2010*). We find that marker genes that are TFs have the highest concordance between species and that the two macaque species, which are also phylogenetically most similar, are also most similar in their ranked marker gene lists. In contrast, lncRNA markers show the lowest overlap between

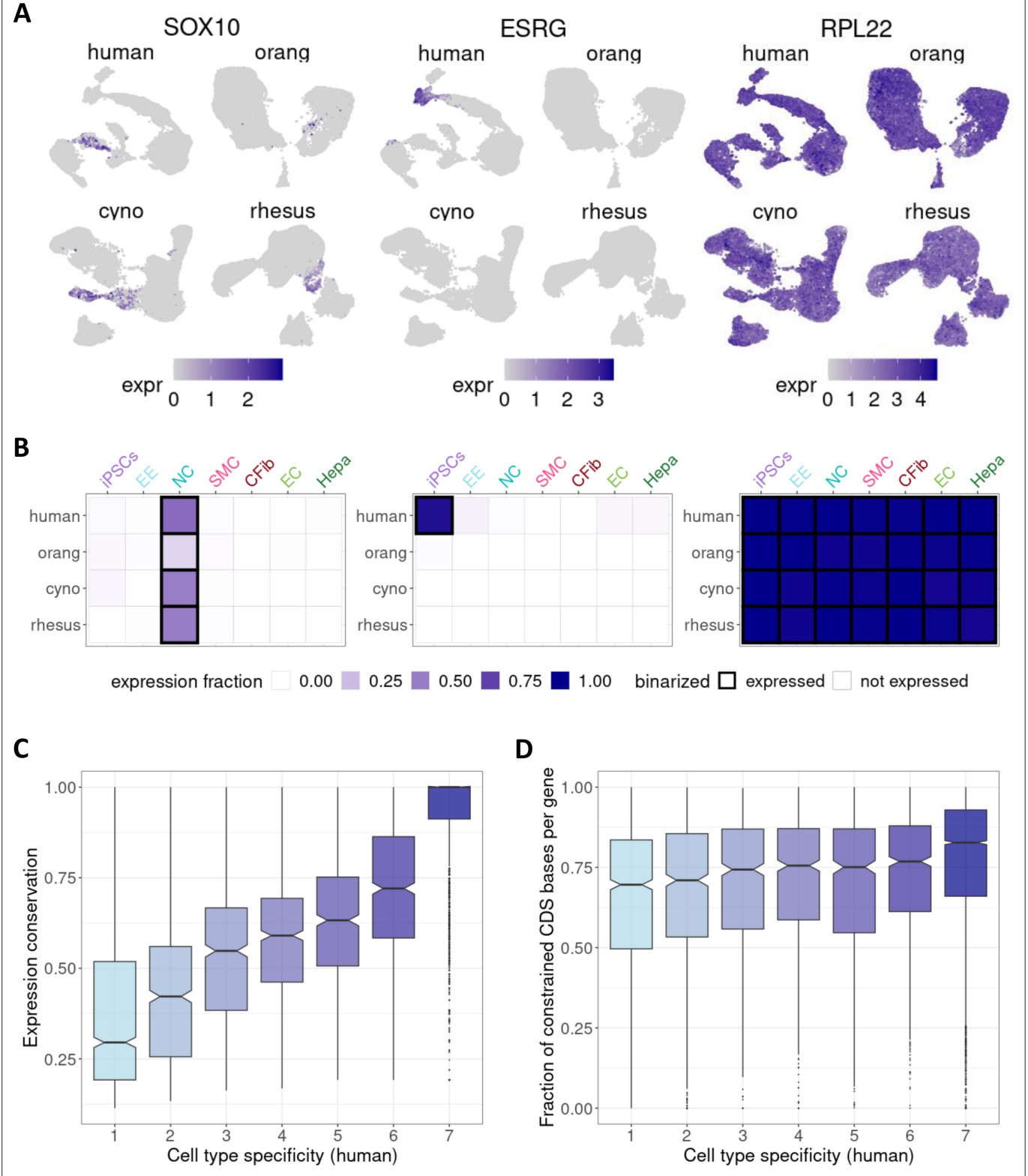

**Figure 3.** Effect of cell type specificity on expression conservation. (**A**) UMAP visualizations depicting expression patterns of selected example genes: *SOX10* (conserved cell type-specific expression in neural crest cells), *ESRG* (species-specific and cell type-specific expression in human iPSCs), and *RPL22* (conserved, broad expression). (**B**) For each gene, expression was summarized per species and cell type as the expression fraction and binarized into 'not expressed'/'expressed' (black frame) based on cell type-specific thresholds. The same example genes as in (**A**) are shown here. iPSCs: induced pluripotent stem cells, EE: early ectoderm, NC: neural crest, SMC: smooth muscle cells, CFib: cardiac fibroblasts, EC: epithelial cells, Hepa: hepatocytes.

*Figure 3 continued on next page*

*Figure 3 continued*

(**C**) Boxplot of expression conservation of genes according to the number of different cell types in which a gene is expressed in humans (cell type specificity). (**D**) Boxplot of the fraction of coding sequence sites that were found to evolve under constraint based on a 43 primate phylogeny (*Sullivan et al., 2023*), stratified by human cell type specificity.

The online version of this article includes the following figure supplement(s) for figure 3:

**Figure supplement 1.** Characteristics of genes with different levels of cell type-specific expression.

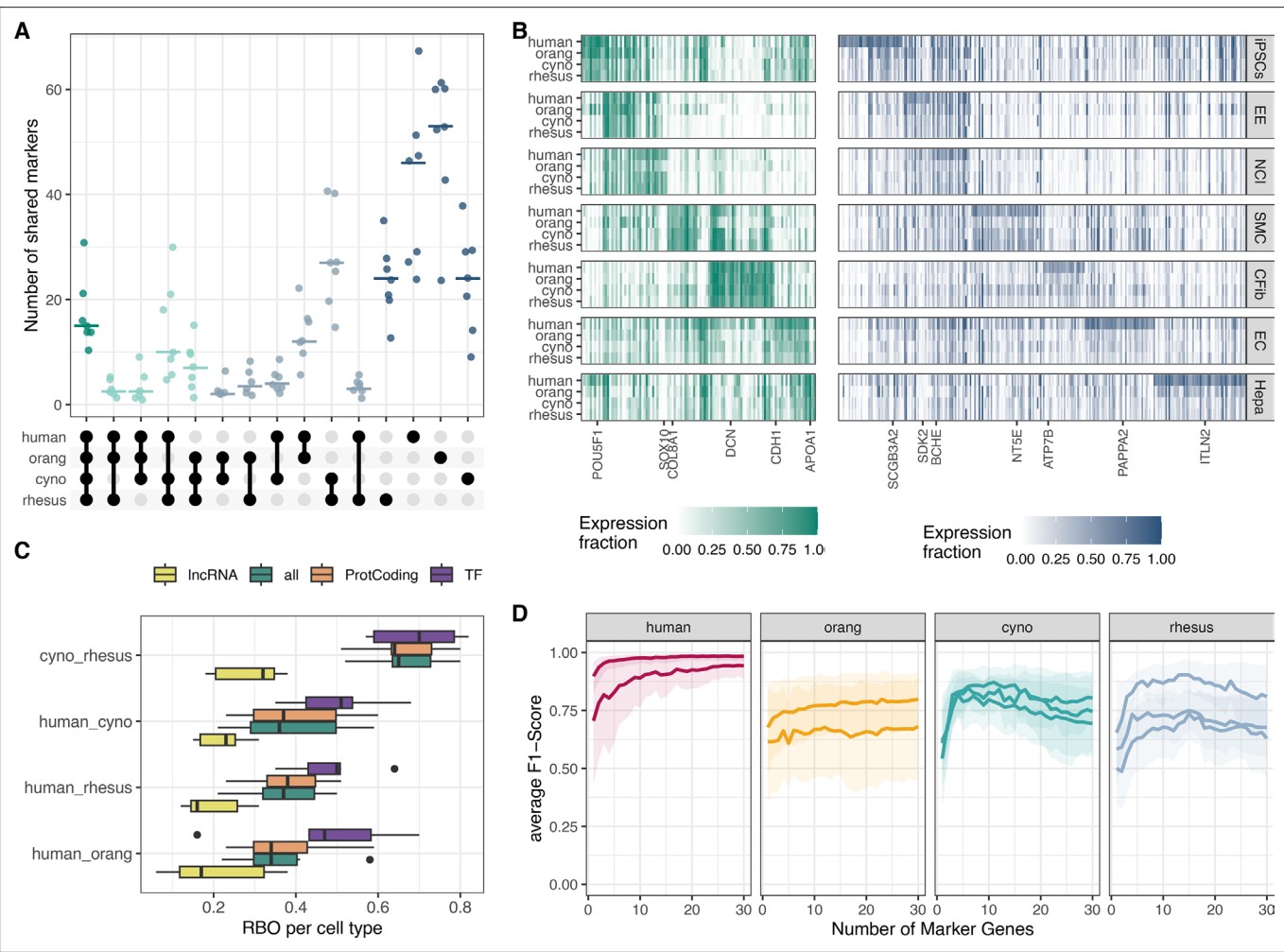

**Figure 4.** Evaluation of marker gene conservation. (**A**) UpSet plot illustrating the overlap between species for the top 100 marker genes per cell type. (**B**) Heatmap showing the expression fractions of marker genes: on the left, markers shared among all species, and on the right, markers unique to the human ranking. For each cell type, one representative gene is labeled and further detailed in *Figure 4—figure supplement 1*. iPSCs: induced pluripotent stem cells, EE: early ectoderm, NC: neural crest, SMC: smooth muscle cells, CFib: cardiac fibroblasts, EC: epithelial cells, Hepa: hepatocytes. (**C**) Rank-biased overlap (RBO) analysis comparing the concordance of gene rankings per cell type for lncRNAs, protein-coding genes, and transcription factors. (**D**) Average F1-score for a k-nearest neighbor (kNN)-classifier trained in the human clone 29B5 to predict cell type identity based on the expression of 1–30 marker genes. Each line represents the performance in a different clone, with shaded areas indicating 95% bootstrap confidence intervals.

The online version of this article includes the following figure supplement(s) for figure 4:

**Figure supplement 1.** Expression patterns of shared and human-specific marker genes.

**Figure supplement 2.** k-nearest neighbor (kNN) classification performance per cell type.

**Figure supplement 3.** k-nearest neighbor (kNN) classification performance for transcription factors and protein coding marker genes.

species. In fact, their cross-species conservation is so low that they also significantly reduce the performance if they are included together with protein-coding markers (*Figure 4C*).

To properly evaluate the performance of marker genes, it is essential to consider their ability to differentiate between cell types. This discriminatory power ultimately determines how well marker genes perform in cell type classification within and across species. To this end, we trained a k-nearest neighbors (kNN) classifier on varying numbers of marker genes per cell type in one human clone (29B5) and evaluated prediction performance using the average F1 score across cell types (*Figure 4—figure supplement 2*). Again, we analyzed markers from a set of all protein-coding genes and TFs only and found that even though TFs appear to be more conserved across species, they do not discriminate cell types as well as the top protein-coding markers (*Figure 4—figure supplement 3*). Using protein-coding marker genes only determined with 29B5 to classify the other human clone, we achieve good discriminatory power (F1 score>0.9) with only 11 marker genes per cell type. In contrast, the classification performance for clones from the other species was substantially lower, failing to reach the performance levels observed in human clones even when using up to 30 marker genes (*Figure 4D*).

In summary, we find that lncRNA marker genes have low transferability between species, while protein-coding markers do reasonably well. However, the predictive value of marker genes decreases with increasing phylogenetic distance, requiring longer marker gene lists to achieve accurate cell type classification for more distantly related species.

## Discussion

An essential criterion for a true cell type is reproducibility across experiments, individuals, or even species. This raises the question of how to reliably identify reproducible cell types across species. When cell types are annotated separately for each species, their reproducibility can be evaluated based on transcriptomic similarity (*Crow et al., 2018*; *Wang et al., 2021*). If integration-based methods are used to accomplish this task (*Barr et al., 2023*; *Bakken et al., 2021*), reproducibility not only depends on the similarity of the expression profiles but also on cell type composition. Integration works best when the cell type compositions are as similar as possible across experiments. This, however, is not the case for organoids, which often have highly heterogeneous cell type compositions (*He et al., 2023*) and our EB data are no exception. Moreover, integration methods struggle with large and variable batch effects, which are expected due to the varying phylogenetic distances across species (*Song et al., 2023*). In contrast, classification methods, such as SingleR (*Aran et al., 2019*) rely mainly on the similarity to a reference profile, which makes it less vulnerable to cell type composition and batch effects. Hence, in our pipeline to identify orthologous cell types, we mainly rely on classification. We start with an unsupervised approach in that we identify cell clusters and then ensure reproducibility as well as comparability using a supervised approach with reciprocal classification of clusters across all species pairs.

Defining cell types in a developmental dataset is particularly challenging, and we do not believe that there is one perfect solution that would fit all cell types and samples. Therefore, we rely on an interactive approach that we implemented in a shiny app (https://shiny.bio.lmu.de/Cross_Species_CellType/) to facilitate the flexible choice of parameters for cluster matching, merging and inspection by visualizing marker genes. *Suresh et al., 2023* employed a similar approach also requiring several manual parameter choices. This makes a formal comparison difficult. Generally, both methods seem to agree well on the orthology assignments of cell type clusters (*Figure 2—figure supplement 2* & *Figure 2—figure supplement 1*). MetaNeighbor, as used by *Suresh et al., 2023*, provides a more quantitative and potentially more sensitive framework for assessing cross-species cell type relationships. However, this higher sensitivity may also make it more affected by data with a lower signal-to-noise ratio, such as our developmental time series.

Hence, the carefully annotated dataset presented here can serve as a valuable resource for future research. Non-human primate iPSCs are central to many studies focusing on evolutionary comparisons, and the pool of iPSC lines for these purposes is expected to grow, incorporating more species and individuals. In this context, the transcriptomic data we generated offer a reference dataset that can be used to verify the pluripotency and differentiation potential of non-human primate iPSC lines by examining gene expression during EB formation.

The set of shared cell types between all four primate species allowed us to evaluate the conservation and transferability of marker genes between species. To begin with, marker genes are by

definition cell type-specific, and also with this dataset, we can show that they are less conserved than broadly expressed genes. Expression breadth can be interpreted as a sign of pleiotropy and hence higher functional constraint (*Hastings, 1996*; *Duret and Mouchiroud, 2000*). Conversely, we expect cell type-specific marker genes to be among the least conserved genes. Indeed, we and others find that the overlap of marker genes across species is limited (*Hodge et al., 2019*; *Krienen et al., 2020*; *Bakken et al., 2021*; *Feng et al., 2022*). Moreover, conservation varies significantly across gene biotypes. On the one hand, lncRNAs, which are often highly cell type-specific, exhibit lower cross-species conservation. Their low sequence conservation further complicates their utility for comparative studies (*Johnsson et al., 2014*). On the other hand, TFs, which have been proposed as central elements of a Core Regulatory Complex (CoRC) that defines cell type identity (*Arendt et al., 2016*), are among the most conserved markers across species. However, the power to distinguish cell types based solely on the expression of TF markers remains lower than when markers are selected from the broader set of all protein-coding genes (*Figure 4—figure supplement 3*). Even though within species, a handful of marker genes can achieve remarkable accuracy, their discriminatory power remains lower for other species. Thus, whole transcriptome profiles offer a more comprehensive approach to cross-species cell type classification for single-cell data.

This said, marker genes remain fundamental to most current cell type annotations. Moreover, marker genes will continue to be used to match cell types across modalities, as, for example, to validate cell type properties in experiments that are often based on immunofluorescence of individual markers or gene panels as used for spatial transcriptomics (*Benito-Kwiecinski et al., 2021*; *Gulati et al., 2025*). To this end, we have refined the ranking of marker genes beyond differential expression analysis to focus on consistent differences in detection rate. Markers identified in this way are bound to translate better into protein-based validations than markers defined based on expression levels, due to the discrepancy of mRNA and protein expression (*Pascal et al., 2008*). Furthermore, the presence-absence signal is more robust against cross-species fluctuations in gene expression than measures based on expression level differences.

In conclusion, we present a robust reference dataset for early primate development alongside tools to identify and evaluate orthologous cell types. Our findings emphasize the need for caution when transferring marker genes for cell type annotation and characterization in cross-species studies.

## Materials and methods
### Cell lines

We used 10 iPSC lines that were all generated in-house and have already been published (*Table 1*). Absence of Sendai virus was confirmed by RT-PCR, and all lines are mycoplasma-free. Cell lines were authenticated using SNP panels that were established using RNA-seq data (*Jocher et al., 2024*).

**Table 1.** Cell lines.
List of cell lines used for embryoid body (EB) differentiation.

| ID | Species | Sex | Publication |
|---|---|---|---|
| 29B5 | *Homo sapiens* | Male | *Geuder et al., 2021* |
| 63Ab2.2 | *Homo sapiens* | Female | *Geuder et al., 2021* |
| 69A1 | *Pongo abelii* | Male | *Geuder et al., 2021* |
| 68A20 | *Pongo abelii* | Male | *Geuder et al., 2021* |
| 82A3 | *Macaca fascicularis* | Female | *Edenhofer et al., 2024* |
| 56B1 | *Macaca fascicularis* | Female | *Edenhofer et al., 2024* |
| 56A1 | *Macaca fascicularis* | Female | |
| 87B1 | *Macaca mulatta* | Male | *Jocher et al., 2024* |
| 83D1 | *Macaca mulatta* | Male | *Jocher et al., 2024* |
| 83Ab1.1 | *Macaca mulatta* | Male | *Jocher et al., 2024* |

## EB differentiation method comparison

Four EB differentiation protocols are compared initially, which are combinations of two differentiation media (DFK20 and EB-medium) and two differentiation methods (dish and 96-well).

For single-cell differentiation in 96-well plates, primate iPSCs from one 80% confluent 6-well are washed with DPBS and incubated with Accumax (Sigma-Aldrich, SCR006) for 7 min at 37 °C. Afterwards, iPSCs are dissociated to single cells, the enzymatic reaction is stopped by adding DPBS, and cells are counted and pelleted at 300×g for 5 min. Single cells are resuspended in EB-medium consisting of StemFit Basic02 (Nippon Genetics, 3821.00) w/o bFGF or DFK20, both supplemented with 10 µM Y-27632 (Biozol, ESI-ST10019). The DFK20 medium consists of DMEM/F12 (Fisher Scientific, 15373541) with 20% KSR (Thermo Fisher Scientific, 10828–028), 1% MEM non-essential amino acids (Thermo Fisher Scientific, 11140–035), 1% Glutamax (Thermo Fisher Scientific, 35050038), 100 U/mL Penicillin, 100 µg/mL Streptomycin (Thermo Fisher Scientific, 15140122), and 0.1 mM 2-Mercaptoethanol (Thermo Fisher Scientific, M3148). Afterwards, 9000 cells in 150 µl medium are seeded per well of a Nuclon Sphera 96-well plate (Fisher Scientific, 15396123) and cultured at 37 °C and 5% $CO_2$. A medium change with the corresponding EB differentiation medium w/o Rock inhibitor is performed every other day during the whole protocol. EBs are collected from the 96-well plate and subjected to flow cytometry after 7 days of differentiation.

For clump differentiation in culture dishes, primate iPSCs from one 80% confluent 12-well are washed with DPBS and incubated with 0.5 mM EDTA (Carl Roth, CN06.3) for 3–5 min at RT. The EDTA is removed, StemFit (Nippon Genetics, 3821.00) supplemented with 10 µM Y-27632 (Biozol, ESI-ST10019) is added and cells are dissociated to clumps of varying sizes. Subsequently, the clumps are transferred to sterile bacterial dishes with vents and cultured at 37 °C and 5% $CO_2$. After 24 hr, the medium is exchanged by either EB-medium or DFK20 supplemented with 10 µM Y-27632 for an additional 24 hr, before changing the medium to EB-medium or DFK20. A medium change is performed every other day during the protocol from day 4 on. EBs are collected from the dishes and subjected to flow cytometry after 7 days of differentiation.

## Flow cytometry

Flow cytometry is performed on day 7 of the differentiation protocol. Therefore, 1/10 of the EBs are collected, washed with DPBS, incubated with Accumax (Sigma-Aldrich, SCR006) for 10 min at 37 °C and dissociated to single cells. After washing, cells are incubated with the Viability Dye eFluor 780 (Thermo Fisher Scientific, 65-0865-18) diluted 1/1000 in PBS for 30 min at 4 °C in the dark. The live/dead stain is quenched by the addition of Cell Staining Buffer (CSB) consisting of DPBS with 0.5% BSA (Sigma-Aldrich, A3059), 0.01% $NaN_3$ (Sigma-Aldrich, S2002), and 2 mM EDTA (Carl Roth, CN06.3). Subsequently, cells are pelleted and incubated with a mixture of the following antibodies diluted 1/200 in CSB for 1 hr at 4 °C in the dark. The antibodies used are anti-TRA-1–60-AF488 (STEMCELL Technologies, 60064AD.1), anti-CXCR4-PE (BioLegend, 306505), anti-NCAM1-PE/Cy7 (BioLegend, 318317), and anti-PDGFRα-APC (BioLegend, 323511). After centrifugation, cells are resuspended in PBS containing 0.5% BSA, 0.01% $NaN_3$, and 1 µg/ml DNase I (STEMCELL Technologies, 07469), filtered through a strainer and analyzed using the BD FACS Canto Flow Cytometry System. Flow cytometry data are analyzed using FlowJo (V10.8.2).

## In-vitro embryoid body differentiation

Two human, two orangutan, three cynomolgus, and three rhesus iPSC lines are used for EB differentiation. The human and orangutan iPSCs are reprogrammed from urinary cells, while cynomolgus and rhesus iPSCs were reprogrammed from fibroblasts. All cell lines were characterized and validated previously and were tested negative for mycoplasma and SeV reprogramming vector integration (*Geuder et al., 2021*; *Jocher et al., 2024*; *Edenhofer et al., 2024*).

For embryoid body formation prior to 10 x scRNA-seq, the EB differentiation protocol using DFK20 medium in culture dishes is performed in duplicates for each clone. After 8 days of floating culture in dishes, EBs from both replicates are pooled and seeded into 6-wells coated with 0.2% gelatin (Sigma-Aldrich, G1890) for another 8 days of attached culture with subsequent medium changes every other day. In total, three replicates of EB formation are performed on different days, and each replicate includes cell lines from all four primate species.

## scRNA-seq library generation and sequencing

EBs are sampled on day 8 and day 16 of the protocol. For dissociation, floating EBs are collected, while attached EBs are kept in their wells, washed with DPBS, and incubated with Accumax (Sigma-Aldrich, SCR006) for 10–20 min at 37 °C. Afterwards, EBs are pipetted up and down with a p1000 pipette until they are completely dissociated. The enzymatic reaction is stopped by adding DFK20 medium, cells are pelleted at 300 g for 5 min and resuspended in 1 mL DPBS. If cell clumps are observed, the liquid is filtered through a 40 μm strainer before counting them with a Countess II automated cell counter (Thermo Fisher Scientific, C10228). Equal cell numbers from each cell line are pooled, washed with DPBS +0.04% BSA and resuspended in DPBS +0.04% BSA aiming for a final concentration of 800–1000 cells/μL. scRNA-seq libraries are generated using the 10 x Genomics Chromium Next GEM Single Cell 3' Kit V3.1 workflow in three replicates. Each time, evenly pooled single cells from the different cell lines are loaded on 2–6 lanes of a 10 x chip, targeting 16,000 cells per lane. Libraries are sequenced on an Illumina NextSeq1000/1500 with a 100-cycle kit and the following sequencing setup: read 1 (28 bases), read 2 (10 bases), read 3 (10 bases), and read 4 (90 bases).

## Alignment of scRNA-seq data

Reads are processed with Cell Ranger version 7.0.0. We map all reads to four reference genomes: *Homo sapiens* GRCh38 (GENCODE release 32), *Pongo abelii* Susie_PABv2/ponAbe3, *Macaca fascicularis* macFas6, and *Macaca mulatta* rheMac10. The orangutan, cynomolgus macaque, and rhesus macaque GTF files are created by transferring the hg38 annotation to the corresponding primate genomes via the tool Liftoff (*Shumate and Salzberg, 2021*), followed by removal of transcripts with partial mapping (<50%), low sequence identity (<50%), or excessive length (>100 bp difference and >2 length ratio) for all species.

## Species and individual demultiplexing

Since we pool cells from multiple species on each 10 x lane, we use cellsnp-lite (*Huang and Huang, 2021*) version 1.2.0 and vireo (*Huang et al., 2019*) version 0.5.7 to assign single cells to their respective species. Initially, we obtain a list of 51000 informative variants (referred to as 'species vcf file') from a bulk RNA-seq experiment involving samples from *Homo sapiens*, *Pongo abelii* and *Macaca fascicularis*, mapped to the GRCh38 reference genome. We run cellsnp-lite in mode 2b for whole-chromosome pileup and filter for high-coverage homozygous variants to identify informative variants.

For the demultiplexing of species in the scRNA-seq data, we employ a two-step strategy:

1. Initial species assignment: Using the Cell Ranger output aligned to GRCh38, we genotype each single cell with cellsnp-lite providing the species vcf file as candidate SNPs and setting a minimum UMI count filter of 10. Subsequently, we assign single cells to human, orangutan, or macaque identity with vireo using again the species vcf file as the donor file.
2. Distinguishing macaque species: To differentiate between the two macaque species, *Macaca fascicularis* and *Macaca mulatta*, we use the Cell Ranger output aligned to rheMac10. After genotyping with cellsnp-lite, we demultiplex with vireo, specifying the number of donors to two, without providing a donor vcf file in this case. We assign the donor, for which the majority of distinguishing variants agreed with the rheMac10 reference alleles, to *Macaca mulatta,* and the other donor to *Macaca fascicularis*.

To distinguish different human individuals pooled in the same experiment, we genotype single cells with cellsnp-lite with a candidate vcf file of 7.4 million common variants from the 1000 Genomes Project, demultiplexed with vireo specifying two donors and assign donors to individuals based on the intersection with variants from bulk RNA-seq data of the same individuals. To distinguish between different cynomolgus individuals, we use a reference vcf with informative variants obtained from bulk RNA-seq data to genotype single cells and demultiplex the individuals.

## Processing of scRNA-seq data

We remove background RNA with CellBender version 0.2.0 (*Fleming et al., 2023*) at a false positive rate (FPR) of 0.01. After quality control, we retain cells with more than 1000 detected genes and a mitochondrial fraction below 8%. We remove cross-species doublets based on the vireo assignments and intra-species doublets using scDblFinder version 1.6.0 (*Germain et al., 2021*), specifying the expected doublet rate based on the cross-species doublet fraction. For each species, we normalize

the counts with scran version 1.28.2 (*Lun et al., 2016*) and integrated data from different experiments with scanorama (*Hie et al., 2019*). UMAP dimensionality reductions are created with Seurat version 4.3.0 on the first 30 components of the scanorama corrected embedding per species.

Besides the separate processing per species, we also create an integrated dataset of all four species together using Harmony version 0.1.1 (*Korsunsky et al., 2019*). We identify clusters on the first 20 Harmony-integrated PCs with Seurat at a resolution of 0.1, resulting in a number of clusters similar to the broad cell types described in a human EB dataset (*Rhodes et al., 2022*; *Figure 1D and E*).

## Reference-based classification

To get an initial cell type annotation, we download a reference dataset of day 21 human EBs (*Rhodes et al., 2022*). We normalize the count matrix with scran and intersect the genes between reference 441 and our scRNA-seq dataset. Next, we train a SingleR version 2.0.0 (*Aran et al., 2019*) classifier for 442 the broad cell type classes defined in *Figure 1G* of the original publication (*Rhodes et al., 2022*) using 443 trainSingleR with pseudo-bulk aggregation. Cell type labels are transferred to cells of each species 444 with *classifySingleR*.

## Orthologous cell type annotation

To annotate orthologous cell types, we first perform high-resolution clustering of the scRNA-seq data for each species separately. For this, we take the first 20 components of the Scanorama-corrected embedding as input to perform clustering in Seurat with *FindNeighbors* and *FindClusters* at a resolution of 2 to obtain the initial HRCs.

Next, we score the similarity of all HRCs with an approach based on reciprocal classification. For each species, we train a SingleR classifier on all HRCs of a species. We then classify the cells of all other species with *classifySingleR*. In this way, we can calculate the similarity of each HRC in the target species to each HRC in the reference species as the fraction of cells of the target HRC classified as the reference HRC. To also obtain similarity scores between HRCs within a species, we split the data of each species into a reference set with 80% of cells and a test set with 20% of cells. Analogous to the cross-species classification scheme, we transfer HRC labels from the reference set to the test set and score the overlap of target and reference HRCs.

In the next step, we combine HRCs based on pairwise similarity scores. We average the bidirectional similarity scores for each HRC pair and construct a distance matrix with all HRCs. Subsequently, based on hierarchical clustering (hclust, average method), we define 26 initial orthologous cell type clusters (OCCs) based on the visual inspection of the distance matrix. In this way, we merge similar HRCs within species and match HRCs across species to obtain a set of OCCs.

OCCs with very similar expression and marker profiles can be further merged. Therefore, we create pseudobulk profiles for each OCC and calculate Spearman's $\rho$ for all pair-wise comparisons within a species (s) based on the 2000 most variable genes. We perform hierarchical clustering on $1 - \bar{\rho}_s$ and merge orthologous clusters at a cut height of 0.1, that was interactively determined by also inspecting the similarity of the top marker genes as found by Seurat's *FindMarkers*. In the shiny app, we provide a list of OCC markers for each species separately, but also the intersection of conserved markers. Based on those marker combinations, the user can then assign the cell types. If the marker gene distribution as visualized in UMAPs reveals overmerged OCCs, the user can split them interactively. Specifically, we separate merged OCC 4 into iPSCs, cardiac progenitor cells and early epithelial cells for the final assignment. We assign merged OCC5 as neural crest I, but re-annotate a subcluster present only in cynomolgus and rhesus macaques as fibroblasts. Similarly, we re-annotate a subcluster of merged OCC12 (granule precursor cells) as astrocyte progenitors in cynomolgus and rhesus macaque. Finally, we exclude OCCs with less than 800 cells that are only present in 1 or 2 species.

We assess the correspondence of the final cell type assignments across species with two approaches. For the scores shown in *Figure 2—figure supplement 1*, we apply the same reciprocal classification approach as described above, providing cell type labels instead of hrcs as initial clusters. For the scores shown in *Figure 2—figure supplement 2*, we use the function MetaNeighborUS of Meta-Neighbor Version 1.18.0 to compare cell type labels across species.

## Pseudotime analysis

Pseudotime trajectories were inferred separately for ectodermal, mesodermal, and endodermal lineages in each species using slingshot (version 2.12.0) (*Street et al., 2018*). For each germ layer, cells were filtered to include iPSCs and cell types belonging to the respective germ layer. The analysis was based on Scanorama-integrated PCA embeddings (*Hie et al., 2019*), with iPSCs defined as the starting cluster and germ layer-specific differentiated cell types as endpoints (ectoderm: astrocyte progenitors, granule precursor cells, neurons, and neural crest cells; mesoderm: fibroblasts, smooth muscle cells, cardiac endothelial cells, and cardiac fibroblasts; endoderm: epithelial cells and hepatocytes). If neural_crest_II was absent, neural_crest_I was used as an alternative endpoint. PHATE embeddings (phateR, version 1.0.7) were (*Moon et al., 2019*) computed from the Scanorama PCA space to visualize the inferred lineages in two dimensions.

## Presence-absence scoring of expression

To determine when to define a gene as expressed in a certain cell type, we derive a lower limit of gene detection per cell type and species while accounting for noise and differences in power to detect expression. We first filter the count matrices for each clone, keeping only genes with at least 1% nonzero counts and cells within three median absolute deviations for number of UMIs and the number of genes with counts >0 per cell type and species. These filtered matrices are then downsampled so that we keep the same number of cells in each species (n=18,800), while keeping the original cell type proportion. Next, per species, we estimate the following distributional characteristics per gene (i) across cell types (j): (1) the fraction of nonzero counts ($f_{ij}$), (2) the mean ($\mu\_ij \pm s.e.(\mu_{ij})$) and dispersion ($\theta\_i$) of the negative binomial distribution using glmgampoi v1.10.2 (*Ahlmann-Eltze and Huber, 2021*). In the next step, we define a putative expression status per gene per cell type. (1) Genes are detectable if their log mean expression $log(\mu_{ij})$ is above the fifth quantile of the $log(\mu)$ value distribution across all genes per cell type. (2) Genes are reliably estimable if the ratio $log(\frac{s.e.(\mu_{ij})}{\mu_{ij}})$ is below the 90th quantile of $log(\frac{s.e.(\mu)}{\mu})$ value distribution. Only when both conditions are met is the expression status set to 1, otherwise 0. A binomial logistic regression model using Firth's bias reduction method as implemented in R package logistf (version 1.26.0) is then applied to derive the minimal gene detection needed to call a gene expressed, i.e., when P(Y=1) solve $log(\frac{p}{1-p}) = a + b * f_{ij}$ towards $f_{ij}$. To ensure consistency between species, we set the detection threshold for each cell type to the maximum threshold among all species.

## Cell type specificity and expression conservation scores

To assess cell type specificity and expression conservation of genes across species, we first determine in which cell types a gene is expressed in a species, using the thresholds defined in the previous section. Thus, we determine cell type specificity as the number of cell types in which a gene was found to be expressed. Here, this score can be maximally 7, i.e., the gene is detected in all cell types that were found in all four species.

To evaluate expression conservation, we develop a phylogenetically weighted conservation score for each gene, reflecting the number of species in which the gene is expressed, weighted by the scaled phylogenetic distance as estimated in *Bininda-Emonds et al., 2007*. For each gene, we calculate the expression conservation score as follows:

$$Expression\ conservation = \frac{1}{N_{ct}} \sum_{ct} \sum_{b \in detected} bl \qquad (1)$$

where $N_{ct}$ is the number of cell types in which the gene is detected. We then simply sum the scaled branch lengths $bl$ across all cell types ($ct$) and branches ($b$) on which we infer the gene to be expressed. Because we only have four species, we only have one internal branch, for which we infer expression if at least one great ape and one macaque species show expression in that cell type. The score ranges from 0.075 (detected only in cynomolgus or rhesus macaque) to 1 (detected in the same cell types in all four species).

Furthermore, we extract measures of sequence conservation for protein-coding genes from Supplementary Data S14 in the study by 2023 (*Sullivan et al., 2023*). Here, we use the fraction of CDS bases with primate phastCons ≥0.96 as a gene-based measure of constraint.

## Marker gene detection

We filter the count matrices for each clone to retain only genes with nonzero counts in one of the 7 cell types that were detected in all species. We then downsample these filtered matrices to equalize the number of cells across species, leaving us with ~11,600 cells per species. Furthermore, to mitigate differences in statistical power due to varying numbers of cells per cell type, we perform testing on cell types with a minimum of 10 and a maximum of 250 cells for each pairwise comparison of 'self' versus 'other.' The maximum of 250 cells ensures that the cell type composition of the 'other' is comparable across species. We identify marker genes using the p-values ($p_{adj} < 0.1$) determined by ZIQ-Rank (*Ling et al., 2021*) and use Seurat *FindMarkers* with logistic regression to identify the cell types for which the gene is a marker. Furthermore, the marker gene needs to be above the cell type's detection threshold (see above) and needs to be up-regulated in the cell type for which it is a marker (log fold change >0.01). Finally, a marker gene must be detected in a larger proportion of cells for which it is a marker than in other cell types ($p_j - \bar{p}_{other} = \Delta > 0.01$). The detection proportion $\Delta$ is also used to sort the lists of marker genes, deeming the genes with the largest $\Delta$ as the best marker genes. In order to also gauge within-species variation in marker gene detection, we conducted the same analysis across clones instead of species. In order to compare cross-species reproducibility of different types of marker genes, i.e., protein-coding, lncRNAs and transcriptional regulators, we wanted to compare the ranked lists of marker genes across species. To this end, we perform a concordance analysis using RBO (*Webber et al., 2010*) on the top 100 marker genes (rbo R package version 0.0.1). For this part, a list of transcription factors were created by selecting genes with at least one annotated motif in the motif databases JASPAR 2022 vertebrate core (*Castro-Mondragon et al., 2022*), JASPAR 2022 vertebrate unvalidated (*Castro-Mondragon et al., 2022*) and IMAGE *Madsen et al., 2018*. Annotations for protein-coding and lncRNA genes were extracted from the Ensembl GTF file provided with the human Cell Ranger reference dataset (GRCh38-2020-A). To assess the predictive performance of marker genes, we conduct a kNN classification (FNN R package version 1.1.4.1). We train a kNN classifier (k=3) on the log-normalized counts of the top 1–30 human markers per cell type in the human clone 29B5. We then predict the cell type identity in all clones and summarize classification performance per cell type with F1-scores, as well as the average F1-score across all seven cell types.

## Acknowledgements

We thank all members of the Enard/Hellmann group for valuable input and discussions. We are grateful to Stefanie Färberböck for her expert technical assistance and help in cell culture. We acknowledge the Core Facility Flow Cytometry at the Biomedical Center, Ludwig-Maximilians-Universität München, for providing equipment and services. We thank Dr. Stefan Krebs and the staff of LAFUGA and the NGS Competence Center Tübingen (NCCT) for sequencing services. This work was supported by the Deutsche Forschungsgemeinschaft (DFG): PJ and JJ, as well as the majority of the project costs, were funded by a grant to IH and WE (458247426). BV was funded by the grant to IH (407541155) and FE by a grant to WE (458888224).

## Additional information

### Funding

| Funder | Grant reference number | Author |
|---|---|---|
| Deutsche Forschungsgemeinschaft | 458247426 | Wolfgang Enard Ines Hellmann |
| Deutsche Forschungsgemeinschaft | 458888224 | Wolfgang Enard Ines Hellmann |
| Deutsche Forschungsgemeinschaft | 407541155 | Ines Hellmann |

The funders had no role in study design, data collection and interpretation, or the decision to submit the work for publication.

### Author contributions

Jessica Jocher, Conceptualization, Data curation, Formal analysis, Validation, Investigation, Visualization, Methodology, Writing – original draft, Project administration; Philipp Janssen, Conceptualization, Data curation, Software, Formal analysis, Validation, Investigation, Visualization, Methodology, Writing – original draft, Writing – review and editing; Beate Vieth, Conceptualization, Software, Formal analysis, Supervision, Investigation, Writing – original draft; Fiona C Edenhofer, Data curation, Methodology; Tamina Dietl, Formal analysis, Methodology; Anita Térmeg, Paulina Spurk, Johanna Geuder, Methodology; Wolfgang Enard, Conceptualization, Supervision, Funding acquisition, Project administration; Ines Hellmann, Conceptualization, Supervision, Funding acquisition, Investigation, Methodology, Writing – original draft, Writing – review and editing

### Author ORCIDs

Philipp Janssen  https://orcid.org/0000-0002-3167-7503
Fiona C Edenhofer  https://orcid.org/0000-0001-6983-2938
Tamina Dietl  https://orcid.org/0009-0000-4126-2603
Anita Térmeg  https://orcid.org/0009-0005-8872-9086
Paulina Spurk  https://orcid.org/0000-0001-8682-370X
Wolfgang Enard  https://orcid.org/0000-0002-4056-0550
Ines Hellmann  https://orcid.org/0000-0003-0588-1313

Reviewer #1 (Public review): https://doi.org/10.7554/eLife.105398.3.sa1
Author response https://doi.org/10.7554/eLife.105398.3.sa2

---

# Additional files

### Supplementary files

MDAR checklist

### Data availability

Code for analysis and figures is available on GitHub (https://github.com/Hellmann-Lab/EB-analyses; copy archived at *Janssen, 2024*), and accompanying files are deposited in Zenodo (https://doi.org/10.5281/zenodo.14198850). All sequencing files were deposited in GEO (GSE280441).

The following datasets were generated:

| Author(s) | Year | Dataset title | Dataset URL | Database and Identifier |
|---|---|---|---|---|
| Jocher J, Janssen P, Vieth B, Edenhofer FC, Dietl T, Térmeg A, Geuder J, Enard W, Hellmann I | 2024 | Identification and comparison of orthologous cell types from primate embryoid bodies shows limits of marker gene transferability | https://www.ncbi.nlm.nih.gov/geo/query/acc.cgi?acc=GSE280441 | NCBI Gene Expression Omnibus, GSE280441 |
| Janssen P | 2024 | Identification and comparison of orthologous cell types from primate embryoid bodies shows limits of marker gene transferability | https://doi.org/10.5281/zenodo.14198849 | Zenodo, 10.5281/zenodo.14198849 |

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

# Appendix 1

**Appendix 1—table 1.** Marker genes.

Literature review for marker genes used in human and mouse / rodents to determine a specific cell type.

| Cell type | Marker gene | Used in human | Used in mouse |
|---|---|---|---|
| iPSCs | POU5F1 | *Nguyen et al., 2018* | *Loh et al., 2006* |
| iPSCs | NANOG | *Nguyen et al., 2018* | *Apostolou et al., 2013* |
| iPSCs | L1TD1 | *Närvä et al., 2012* | *Närvä et al., 2012* |
| early ectoderm | SOX2 | *Graham et al., 2003* | *Lodato et al., 2013* |
| early ectoderm | HES5 | *Ziller et al., 2015* | *Harada et al., 2021* |
| early ectoderm | RFX4 | *Ziller et al., 2015* | *Kawase et al., 2014* |
| granule precursor cells | NFIA | *Tan et al., 2023* | *Fraser et al., 2020* |
| granule precursor cells | ZIC1 | *Aruga et al., 1998* | *Schüller et al., 2006* |
| granule precursor cells | ZIC4 | *Aruga et al., 1998* | *Blank et al., 2011* |
| neural crest | SOX10 | *Mollaaghababa and Pavan, 2003* | *Mollaaghababa and Pavan, 2003*; *Kim et al., 2003* |
| neural crest | FOXD3 | *Tseng et al., 2016* | *Dottori et al., 2001* |
| neural crest | S100B | *Hackland et al., 2017* | *Murphy et al., 1991* |
| neurons | STMN2 | *Klim et al., 2019* | *Guerra San Juan et al., 2022*; *Ware et al., 2016* |
| neurons | TAGLN3 (NP25) | *Mori et al., 2004* | *Ware et al., 2016* |
| neurons | DCX | *Gleeson et al., 1999* | *Gleeson et al., 1999* |
| smooth muscle cells | COL8A1 | *Rojas et al., 2024* | *Muhl et al., 2022* |
| smooth muscle cells | ACTG2 | *Hashmi et al., 2020* | *Muhl et al., 2022* |
| smooth muscle cells | ACTA2 | *Rojas et al., 2024* | *Muhl et al., 2022* |
| cardiac fibroblasts | TNNT2 | *Mononen et al., 2020* | *Tachampa and Wongtawan, 2020* |
| cardiac fibroblasts | DCN | *Floy et al., 2021* | *Ko et al., 2022* |
| cardiac fibroblasts | HAND2 | *Mononen et al., 2020* | *Furtado et al., 2014* |
| epithelial cells | CDH1 | *Oikawa et al., 2018* | *Bondow et al., 2012* |
| epithelial cells | EPCAM | *Martowicz et al., 2016* | *Huang et al., 2018* |
| epithelial cells | CLDN7 | *Farkas et al., 2015* | *Xing et al., 2020* |
| hepatocytes | TTR | *Banas et al., 2007* | *Lavon and Benvenisty, 2005* |
| hepatocytes | APOA1 | *Krueger et al., 2013* | *De Giorgi et al., 2021* |
| hepatocytes | APOA2 | *Krueger et al., 2013* | *Peng et al., 2018* |

