## [Editor Report · eLife Assessment]

The authors make an **important** contribution to comparative functional genomics by developing a semi-automated computational pipeline that integrates classification and marker-based cluster annotation to identify orthologous cell types. Using a single-cell RNA-seq dataset of induced pluripotent stem cells and derived embryonic bodies from four primate species: humans, orangutans, cynomolgus macaques, and rhesus macaques, the authors provide **convincing** evidence that cell type-specific marker genes are substantially less transferable across species than broadly expressed genes, with transferability declining as phylogenetic distance increases. This study establishes a key framework and reference dataset for comparative single-cell analyses and encourages more rigorous evaluation of marker gene transferability across species.

---

## [Referee Report · Reviewer #1 (Public review)]

Summary:

Jocher, Janssen et al examine the robustness of comparative functional genomics studies in primates that make use of induced pluripotent stem cell-derived cells. Comparative studies in primates, especially amongst the great apes, are generally hindered by the very limited availability of samples, and iPSCs, which can be maintained in the laboratory indefinitely and defined into other cell types, have emerged as promising model systems because they allow the generation of data from tissues and cells that would otherwise would be unobservable.

Undirected differentation of iPSCs into many cell types at once, using a method known as embryoid body differentiation, requires researchers to manually assign all cell types in the dataset so they can be correctly analysed. Typically, this is done using marker genes associated with a specific cell type. These are defined a priori, and have historically tended to be characterised in mice and human and then employed to annotate other species. Jocher, Janssen et al ask if the marker genes and features used to define a given cell type in one species are suitable for use in a second species, and then quantify the degree of usefulness of these markers. They find that genes that are informative and cell type specific in a given species are less valuable for cell type identification in other species, and that this value, or transferability, drops off as the evolutionary distance between species increases.

This paper will help guide future comparative studies of gene expression in primates (and more broadly) as well as add to the growing literature on the broader challenges of selecting powerful and reliable marker genes for use in single cell transcriptomics.

Strengths:

Marker gene selection and cell type annotation is challenging problem in scRNA studies, and successful classification of cells often requires manual expert input. This can be hard to reproduce across studies, as despite general agreement on the identity of many cell types, different methods for identifying marker genes will return different sets of genes. The rise of comparative functional genomics complicates this even further, as a robust marker gene in one species need not always be as useful in a different taxon. The finding that so many marker genes have poor transferability is striking, and by interrogating the assumption of transferability in a thorough and systematic fashion, this paper reminds us of the importance of systematically validating analytical choices. The focus on identifying how transferability varies across different types of marker genes (especially when comparing TFs to lncRNAs), and on exploring different methods to identify marker genes, also suggests additional criteria by which future researchers could select robust marker genes in their own data.

The paper is built on a substantial amount of clearly reported and thoroughly considered data, including EBs and cells from four different primate species - humans, orangutans, and two macaque species. The authors go to great lengths to ensure the EBs are as comparable as possible across species, and take similar care with their computational analyses, always erring on the side of drawing conservative conclusions that are robustly supported by their data over more tenuously supported ones that could be impacted by data processing artefacts such as differences in mappability etc. For example, I like the approach of using liftoff to robustly identify genes in non-human species that can be mapped to and compared across species confidently, rather than relying on the likely incomplete annotation of the non-human primate genomes. The authors also provide an interactive data visualisation website that allows users to explore the dataset in depth, examine expression patterns of their own favourite marker genes and perform the same kinds of analyses on their own data if desired, facilitating consistency between comparative primate studies.

Weaknesses and recommendations:

(1) Embryoid body generation is known to be highly variable from one replicate to the next for both technical and biological reasons, and the authors do their best to account for this, both by their testing of different ways of generating EBs, and by including multiple technical replicates/clones per species. However, there is still some variability that could be worth exploring in more depth. For example, the orangutan seems to have differentiated preferentially towards cardiac mesoderm whereas the other species seemed to prefer ectoderm fates, as shown in Figure 2C. Likewise, Supplementary Figure 2C suggests significant unbalance in the contributions across replicates within a species, which is not surprising given the nature of EBs, while Supplementary Figure 6 suggests that despite including three different clones from a single rhesus macaque, most of the data came from a single clone. The manuscript would be strengthened by a more thorough exploration of the intra-species patterns of variability, especially for the taxa with multiple biological replicates, and how they impact the number of cell types detected across taxa etc.

The same holds for the temporal aspect of the data, which is not really discussed in depth despite being a strength of the design. Instead, days 8 and 16 are analysed jointly, without much attention being paid to the possible differences between them. Are EBs at day 16 more variable between species than at day 8? Is day 8 too soon to do these kinds of analyses? Are markers for earlier developmental progenitors better/more transferable than those for more derived cell types?

(2) Closely tied to the point above, by necessity the authors collapse their data into seven fairly coarse cell types, and then examine the performance of canonical marker genes (as well as those discovered de novo) across the species. But some of the clusters they use are somewhat broad, and so it is worth asking whether the lack of specificity exhibited by some marker genes and driving their conclusions is driven by inter-species heterogeneity within a given cluster.

Comments on revisions:

I think the authors have addressed my previous comments to my satisfaction, and I thank them for the changes they have made, it's good to see that the manuscript is just as sound as it seemed the first time around.

---

## [Author Response]

The following is the authors’ response to the original reviews.

Most importantly, in accordance with questions raised by Reviewer 1, we now include a detailed comparison of the cell type frequencies between the two examined time points as well as comparison of the pseudotimes along those lineages. This is detailed in the new section “Many cell types are shared between day 8 and day 16 EBs” and illustrated in Supplementary Figure 6c and Supplementary Figures 7-8.

Besides this new chapter and its accompanying methods part, we mainly edited the language and to clarify methods and assumptions according to the Reviewer suggestions.

The main concern of Reviewer 2 was our use of the liftoff gene annotation. We explained our reasoning for this choice extensively in our public response to the Reviewer, but did not incorporate this into our manuscript because even though this is an important subject it is not within the main scope of our paper.

**Public Reviews:**

**Reviewer #1 (Public review):**
Summary:Jocher, Janssen, et al examine the robustness of comparative functional genomics studies in primates that make use of induced pluripotent stem cell-derived cells. Comparative studies in primates, especially amongst the great apes, are generally hindered by the very limited availability of samples, and iPSCs, which can be maintained in the laboratory indefinitely and defined into other cell types, have emerged as promising model systems because they allow the generation of data from tissues and cells that would otherwise be unobservable.Undirected differentiation of iPSCs into many cell types at once, using a method known as embryoid body differentiation, requires researchers to manually assign all cell types in the dataset so they can be correctly analysed. Typically, this is done using marker genes associated with a specific cell type. These are defined a priori, and have historically tended to be characterised in mice and humans and then employed to annotate other species. Jocher, Janssen, et al ask if the marker genes and features used to define a given cell type in one species are suitable for use in a second species, and then quantify the degree of usefulness of these markers. They find that genes that are informative and cell type specific in a given species are less valuable for cell type identification in other species, and that this value, or transferability, drops off as the evolutionary distance between species increases.This paper will help guide future comparative studies of gene expression in primates (and more broadly) as well as add to the growing literature on the broader challenges of selecting powerful and reliable marker genes for use in single-cell transcriptomics.Strengths:Marker gene selection and cell type annotation is a challenging problem in scRNA studies, and successful classification of cells often requires manual expert input. This can be hard to reproduce across studies, as, despite general agreement on the identity of many cell types, different methods for identifying marker genes will return different sets of genes. The rise of comparative functional genomics complicates this even further, as a robust marker gene in one species need not always be as useful in a different taxon. The finding that so many marker genes have poor transferability is striking, and by interrogating the assumption of transferability in a thorough and systematic fashion, this paper reminds us of the importance of systematically validating analytical choices. The focus on identifying how transferability varies across different types of marker genes (especially when comparing TFs to lncRNAs), and on exploring different methods to identify marker genes, also suggests additional criteria by which future researchers could select robust marker genes in their own data.The paper is built on a substantial amount of clearly reported and thoroughly considered data, including EBs and cells from four different primate species - humans, orangutans, and two macaque species. The authors go to great lengths to ensure the EBs are as comparable as possible across species, and take similar care with their computational analyses, always erring on the side of drawing conservative conclusions that are robustly supported by their data over more tenuously supported ones that could be impacted by data processing artefacts such as differences in mappability, etc. For example, I like the approach of using liftoff to robustly identify genes in non-human species that can be mapped to and compared across species confidently, rather than relying on the likely incomplete annotation of the non-human primate genomes. The authors also provide an interactive data visualisation website that allows users to explore the dataset in depth, examine expression patterns of their own favourite marker genes and perform the same kinds of analyses on their own data if desired, facilitating consistency between comparative primate studies.

We thank the Reviewer for their kind assessment of our work.

Weaknesses and recommendations:(1) Embryoid body generation is known to be highly variable from one replicate to the next for both technical and biological reasons, and the authors do their best to account for this, both by their testing of different ways of generating EBs, and by including multiple technical replicates/clones per species. However, there is still some variability that could be worth exploring in more depth. For example, the orangutan seems to have differentiated preferentially towards cardiac mesoderm whereas the other species seemed to prefer ectoderm fates, as shown in Figure 2C. Likewise, Supplementary Figure 2C suggests a significant unbalance in the contributions across replicates within a species, which is not surprising given the nature of EBs, while Supplementary Figure 6 suggests that despite including three different clones from a single rhesus macaque, most of the data came from a single clone. The manuscript would be strengthened by a more thorough exploration of the intra-species patterns of variability, especially for the taxa with multiple biological replicates, and how they impact the number of cell types detected across taxa, etc.

You are absolutely correct in pointing out that the large clonal variability in cell type composition is a challenge for our analysis. We also noted the odd behavior of the orangutan EBs, and their underrepresentation of ectoderm. There are many possible sources for these variable differentiation propensities: clone, sample origin (in this case urine) and individual. However, unfortunately for the orangutan, we have only one individual and one sample origin and thus cannot say whether this germ layer preference says something about the species or is due to our specific sample. Because of this high variability from multiple sources, getting enough cell types with an appreciable overlap between species was limiting to analyses. In order to be able to derive meaningful conclusions from intra-species analyses and the impact of different sources of variation on cell type propensity, we would need to sequence many more EBs with an experimental design that balances possible sources of variation. This would go beyond the scope of this study.

Instead, here we control for intra-species variation in our analyses as much as possible: For the analysis of cell type specificity and conservation the comparison is relative for the different specificity degrees (Figure 3C). For the analysis of marker gene conservation, we explicitly take intra-species variation into account (Figure 4D).

The same holds for the temporal aspect of the data, which is not really discussed in depth despite being a strength of the design. Instead, days 8 and 16 are analysed jointly, without much attention being paid to the possible differences between them.

Concerning the temporal aspect, indeed we knowingly omitted to include an explicit comparison of day 8 and day 16 EBs, because we felt that it was not directly relevant to our main message. Our pseudotime analysis showed that the differences of the two time points were indeed a matter of degree and not so much of quality. All major lineages were already present at day 8 and even though day 8 cells had on average earlier pseudotimes, there was a large overlap in the pseudotime distributions between the two sampling time points (Author response image 1). That is why we decided to analyse the data together.

Are EBs at day 16 more variable between species than at day 8? Is day 8 too soon to do these kinds of analyses?

When we started the experiment, we simply did not know what to expect. We were worried that cell types at day 8 might be too transient, but longer culture can also introduce biases. That is why we wanted to look at two time points, however as mentioned above the differences are in degree.

Concerning the cell type composition: yes, day 16 EBs are more heterogeneous than day 8 EBs. Firstly, older EBs have more distinguishable cell types and hence even if all EBs had identical composition, the sampling variance would be higher given that we sampled a similar number of cells from both time points. Secondly, in order to grow EBs for a longer time, we moved them from floating to attached culture on day 8 and it is unclear how much variance is added by this extra handling step.

Are markers for earlier developmental progenitors better/more transferable than those for more derived cell types?

We did not see any differences in the marker conservation between early and late cell types, but we have too little data to say whether this carries biological meaning.

**Author response image 1. sa2fig1:** Pseudotime analysis for a differentiation trajectory towards neurons. Single cells were first aggregated into metacells per species using SEACells (Persad et al. 2023). Pluripotent and ectoderm metacells were then integrated across all four species using Harmony and a combined pseudotime was inferred with Slingshot (Street et al. 2018), specifying iPSCs as the starting cluster. Here, lineage 3 is shown, illustrating a differentiation towards neurons. (A) PHATE embedding colored by pseudotime (Moon et al. 2019). (B) PHATE embedding colored by celltype. (C) Pseudotime distribution across the sampling timepoints (day 8 and day 16) in different species.

(2) Closely tied to the point above, by necessity the authors collapse their data into seven fairly coarse cell types and then examine the performance of canonical marker genes (as well as those discovered de novo) across the species. However some of the clusters they use are somewhat broad, and so it is worth asking whether the lack of specificity exhibited by some marker genes and driving their conclusions is driven by inter-species heterogeneity within a given cluster.

**Author response image 2. sa2fig2:** UMAP visualization for the Harmony-integrated dataset across all four species for the seven shared cell types, colored by cell type identity (A) and species (B).

Good point, if we understand correctly, the concern is that in our relatively broadly defined cell types, species are not well mixed and that this in turn is partly responsible for marker gene divergence. This problem is indeed difficult to address, because most approaches to evaluate this require integration across species which might lead to questionable results (see our Discussion).

Nevertheless, we attempted an integration across all four species. To this end, we subset the cells for the 7 cell types that we found in all four species and visualized cell types and species in the UMAPs above (Author response image 2).

We see that cardiac fibroblasts appear poorly integrated in the UMAP, but they still have very transferable marker genes across species. We quantified integration quality using the cell-specific mixing score (cms) (Lütge et al. 2021) and indeed found that the proportion of well integrated cells is lowest for cardiac fibroblasts (Author response image 3A). On the other end of the cms spectrum, neural crest cells appear to have the best integration across species, but their marker transferability between species is rather worse than for cardiac fibroblasts (Supplementary Figure 9). Cell-type wise calculated rank-biased overlap scores that we use for marker gene conservation show the same trends (Author response image 3B) as the F1 scores for marker gene transferability. Hence, given our current dataset we do not see any indication that the low marker gene conservation is a result of too broadly defined cell types.

**Author response image 3. sa2fig3:** (A) Evaluation of species mixing per cell type in the Harmony-integrated dataset, quantified by the fraction of cells with an adjusted cell-specific mixing score (cms) above 0. 05. (B) Summary of rank-biased overlap (RBO) scores per cell type to assess concordance of marker gene rankings for all species pairs.

**Reviewer #2 (Public review):**
Summary:The authors present an important study on identifying and comparing orthologous cell types across multiple species. This manuscript focuses on characterizing cell types in embryoid bodies (EBs) derived from induced pluripotent stem cells (iPSCs) of four primate species, humans, orangutans, cynomolgus macaques, and rhesus macaques, providing valuable insights into cross-species comparisons.Strengths:To achieve this, the authors developed a semi-automated computational pipeline that integrates classification and marker-based cluster annotation to identify orthologous cell types across primates. This study makes a significant contribution to the field by advancing cross-species cell type identification.

We thank the reviewer for their positive and thoughtful feedback.

Weaknesses:However, several critical points need to be addressed.(1) Use of Liftoff for GTF AnnotationThe authors used Liftoff to generate GTF files for Pongo abelii, Macaca fascicularis, and *Macaca mulatta* by transferring the hg38 annotation to the corresponding primate genomes. However, it is unclear why they did not use species-specific GTF files, as all these genomes have existing annotations. Why did the authors choose not to follow this approach?

As Reviewer 1 also points out, also we have observed that the annotation of non-human primates often has truncated 3’UTRs. This is especially problematic for 3’ UMI transcriptome data as the ones in the 10x dataset that we present here. To illustrate this we compared the Liftoff annotation derived from Gencode v32, that we also used throughout our manuscript to the Ensembl gene annotation Macaca_fascicularis_6.0.111. We used transcriptomes from human and cynomolgus iPSC bulk RNAseq (Kliesmete et al. 2024) using the Prime-seq protocol (Janjic et al. 2022) which is very similar to 10x in that it also uses 3’ UMIs. On average using Liftoff produces higher counts than the Ensembl annotation (Author response image 4A). Moreover, when comparing across species, using Ensembl for the macaque leads to an asymmetry in differentially expressed genes, with apparently many more up-regulated genes in humans. In contrast, when we use the Liftoff annotation, we detect fewer DE-genes and a similar number of genes is up-regulated in macaques as in humans (Author response image 4B). We think that the many more DE-genes are artifacts due to mismatched annotation in human and cynomolgus macaques. We illustrate this for the case of the transcription factor SALL4 in Author response image 4C, D. The Ensembl annotation reports 2 transcripts, while Liftoff from Gencode v32 suggests 5 transcripts, one of which has a longer 3’UTR. This longer transcript is also supported by Nanopore data from macaque iPSCs. The truncation of the 3’UTR in this case leads to underestimation of the expression of SALL4 in macaques and hence SALL4 is detected as up-regulated in humans (DESeq2: LFC = 1.34, p-adj<2e-9). In contrast, when using the Liftoff annotation SALL4 does not appear to be DE between humans and macaques (LFC=0.33, p.adj=0.20).

**Author response image 4. sa2fig4:** (A) UMI-counts/ gene for the same cynomolgus macaque iPSC samples. On the x-axis the gtf file from Ensembl Macaca_fascicularis_6.0.111 was used to count and on the y-axis we used our filtered Liftoff annotation that transferred the human gene models from Gencode v32. (B) The # of DE-genes between human and cynomolgus iPSCs detected with DESeq2. In Liftoff, we counted human samples using Gencode v32 and compared it to the Liftoff annotation of the same human gene models to macFas6. In Ensembl, we use Gencode v32 for the human and Ensembl Macaca_fascicularis_6.0.111 for the Macaque. For both comparisons we subset the genes to only contain one-to-one orthologs as annotated in biomart. Up and down regulation is relative to human expression. (C) Read counts for one example gene SALL4. Here, we used in addition to the Liftoff and Ensembl annotation also transcripts derived from Nanopore cDNA sequencing of cynomolgus iPSCs. (D) Gene models for SALL4 in the space of MacFas6 and a coverage for iPSC-Prime-seq bulk RNA-sequencing.

(2) Transcript Filtering and Potential BiasesThe authors excluded transcripts with partial mapping (<50%), low sequence identity (<50%), or excessive length differences (>100 bp and >2× length ratio). Such filtering may introduce biases in read alignment. Did the authors evaluate the impact of these filtering choices on alignment rates?

We excluded those transcripts from analysis in both species, because they present a convolution of sequence-annotation differences and expression. The focus in our study is on regulatory evolution and we knowingly omit marker differences that are due to a marker being mutated away, we will make this clearer in the text of a revised version.

(3) Data Integration with HarmonyThe methods section does not specify the parameters used for data integration with Harmony. Including these details would clarify how cross-species integration was performed.

We want to stress that none of our conservation and marker gene analyses relies on cross-species integration. We only used the Harmony integrated data for visualisation in Figure 1 and the rough germ-layer check up in Supplementary Figure S3. We will add a better description in the revised version.

Reference

Janjic, Aleksandar, Lucas E. Wange, Johannes W. Bagnoli, Johanna Geuder, Phong Nguyen, Daniel Richter, Beate Vieth, et al. 2022. “Prime-Seq, Efficient and Powerful Bulk RNA Sequencing.” Genome Biology 23 (1): 88.

Kliesmete, Zane, Peter Orchard, Victor Yan Kin Lee, Johanna Geuder, Simon M. Krauß, Mari Ohnuki, Jessica Jocher, Beate Vieth, Wolfgang Enard, and Ines Hellmann. 2024. “Evidence for Compensatory Evolution within Pleiotropic Regulatory Elements.” Genome Research 34 (10): 1528–39.

Lütge, Almut, Joanna Zyprych-Walczak, Urszula Brykczynska Kunzmann, Helena L. Crowell, Daniela Calini, Dheeraj Malhotra, Charlotte Soneson, and Mark D. Robinson. 2021. “CellMixS: Quantifying and Visualizing Batch Effects in Single-Cell RNA-Seq Data.” Life Science Alliance 4 (6): e202001004.

Moon, Kevin R., David van Dijk, Zheng Wang, Scott Gigante, Daniel B. Burkhardt, William S. Chen, Kristina Yim, et al. 2019. “Visualizing Structure and Transitions in High-Dimensional Biological Data.” Nature Biotechnology 37 (12): 1482–92.

Persad, Sitara, Zi-Ning Choo, Christine Dien, Noor Sohail, Ignas Masilionis, Ronan Chaligné, Tal Nawy, et al. 2023. “SEACells Infers Transcriptional and Epigenomic Cellular States from Single-Cell Genomics Data.” Nature Biotechnology 41 (12): 1746–57.

Street, Kelly, Davide Risso, Russell B. Fletcher, Diya Das, John Ngai, Nir Yosef, Elizabeth Purdom, and Sandrine Dudoit. 2018. “Slingshot: Cell Lineage and Pseudotime Inference for Single-Cell Transcriptomics.” BMC Genomics 19 (1): 477.

**Recommendations for the authors:**

**Reviewer #1 (Recommendations for the authors):**
(1) Figure 1B: the orangutan tubulin stain looks a bit unusual - just confirming that this is indeed the right image the authors want to include here.

We agree, this unfortunately also reflects the findings from the scRNA-seq analysis in that we found hardly any cells that we would classify as proper neurons.

(2) Typo on line 90: 'loosing' should be 'losing'.

Fixed

(3) Line 118: why do the authors believe that using singleR will give better results than MetaNeighbour? This certainly seems supported by the data in S4 and S5, but the reasoning is not clear.

We think that this might depend on the signal to noise ratio, which is a property specific to each dataset. Here we just wanted to state that our approach seems to work better for our developmental data, but we didn’t test out other data and thus cannot generalize.

(4) Figure 2B: there are some coloured lines on the first filled black bar from the left - do they mean anything? I couldn't work it out from looking at the figure.

Indeed this is a bit misleading the colors on the left represent the species identity: this was to illustrate the mixing of the of species for each cell type: The legend reads now: “Each line represents a cell which are colored by their species of origin on the left and by their current cell type assignment during the annotation procedure on the right.”

(5) Figure 3: I did not understand how the seven bins of the cell type specificity metric were derived until much later - it is just the number of cell types in which a gene is expressed, yes? Might be worth making this clearer earlier in the text.

We made this more explicit in the legend. “Boxplot of expression conservation of genes according to the number of different cell types in which a gene is expressed in humans (cell type specificity).”

(6) It would be great to provide a bit more thorough documentation for the shiny app, so it can serve as a stand-alone resource and not require going back and forth with the paper to make sure one knows what one is doing at every point.

Agree, this would be a good idea. We are on it.

(7) Line 477: I think this is unclear - the authors retain over 11000 cells per species but then set the maximum number of cells in a cluster for pairwise comparison to 250... which is a lot fewer. What happens to all the other cells? This probably needs some rewriting to clarify it.

We did this to minimize the power differences due to cell numbers and thus make the results more comparable across species. We added this explanation to the methods section for Marker gene detection.

**Reviewer #2 (Recommendations for the authors):**
How was the clustering resolution (0.1) determined?

This resolution was only used for the initial rough check up of the germ layers as reported in Figure 1 and Supplementary Figures S3. We chose this resolution because it yielded roughly the same number of clusters as the number of cell types that we got from classification with the Rhodes et al data.